# Deep Isolated Aquifer Brines Harbor Atypical Halophilic Microbial Communities in Quebec, Canada

**DOI:** 10.3390/genes14081529

**Published:** 2023-07-26

**Authors:** Jean-Christophe Gagnon, Samuel Beauregard-Tousignant, Jean-Sébastien Marcil, Cassandre Sara Lazar

**Affiliations:** 1Department of Biological Sciences, University of Québec at Montréal (UQAM), C.P. 8888, Succ. Centre-Ville, Montréal, QC H3C 3P8, Canada; pommepoire91@gmail.com (J.-C.G.); beauregard-tousignant.samuel@courrier.uqam.ca (S.B.-T.); 2Interuniversity Research Group in Limnology/Groupe de Recherche Interuniversitaire en Limnologie (GRIL), Montréal, QC H3C 3P8, Canada; 3Derena Geosciences, Quebec, QC G7A 3Y5, Canada; derena@videotron.ca; 4Ressources Utica Inc., Quebec, QC G1V 4M7, Canada

**Keywords:** deep terrestrial subsurface, brine, groundwater bacteria, endolithic bacteria, metagenomics

## Abstract

The deep terrestrial subsurface, hundreds of meters to kilometers below the surface, is characterized by oligotrophic conditions, dark and often anoxic settings, with fluctuating pH, salinity, and water availability. Despite this, microbial populations are detected and active, contributing to biogeochemical cycles over geological time. Because it is extremely difficult to access the deep biosphere, little is known about the identity and metabolisms of these communities, although they likely possess unknown pathways and might interfere with deep waste deposits. Therefore, we analyzed rock and groundwater microbial communities from deep, isolated brine aquifers in two regions dating back to the Ordovician and Devonian, using amplicon and whole genome sequencing. We observed significant differences in diversity and community structure between both regions, suggesting an impact of site age and composition. The deep hypersaline groundwater did not contain typical halophilic bacteria, and genomes suggested pathways involved in protein and hydrocarbon degradation, and carbon fixation. We identified mainly one strategy to cope with osmotic stress: compatible solute uptake and biosynthesis. Finally, we detected many bacteriophage families, potentially indicating that bacteria are infected. However, we also found auxiliary metabolic genes in the viral genomes, probably conferring an advantage to the infected hosts.

## 1. Introduction

The terrestrial subsurface is one of the biggest terrestrial ecosystems on the planet. It spans a few centimeters to kilometers under the surface [1] and is composed of sediments, rocks, gases, and pores through which groundwater flows. It is a dark, oligotrophic environment with varying oxygen, pH, and salinity concentrations, as well as limited energy sources [2]. However, microbial life prevails in these habitats [3], with heterotrophs possibly consuming organic compounds derived from sediments and rocks and lithotrophic bacteria using H_2_ [4]. Furthermore, subsurface microbial communities likely represent 40% of the planet’s microbial biomass [5]. However, the deep biosphere remains poorly understood, mainly because of the difficulties in sample collection. A study from two deep confined aquifers in Washington revealed the presence of oxygen, Fe(III), NO_3_^−^, SO_4_^2−^, or HCO_3_^−^ reducers, heterotroph fermenters, and methanogens [6]. Kadnikov et al. [7] detected hydrogenotrophic methanogens and heterotrophs using proteins as carbon sources in a 2.8 km-deep aquifer in Russia. Probst et al. [8] found CPR (candidate phyla radiation) bacteria in a deep, CO_2_-rich aquifer.

These deep subsurface microbial communities are extremely diverse and cover all three domains of life [2], as well as viruses. Endolithic microbes can colonize rock matrices, limited by rock pore size, temperature, nutrient and energy sources, and the presence of water [4]. Given the unique conditions found in the deep subsurface, it is probable that novel metabolisms to scavenge nutrients and harness energy are employed by the underground microbes. Many concerns actually exist about how these deep microbial communities will react to the digging and use of isolated deep bedrock environments as nuclear and hazardous waste repositories [2,6]. Thus, we first need to understand who lives in the deep biosphere and which biochemical pathways are used by these microbial communities.

Therefore, in this study, we analyzed rock and groundwater hypersaline brine samples from the St. Lawrence lowlands in the Bécancour area and from the Gaspésie area in the Quebec province (Canada). Brines in the Bécancour area date back to the Ordovician era and are composed of Na, Ca, and Cl, possibly derived from halite dissolution [9]. The presence of hypersaline brine in the deep geological reservoirs of Bécancour is possibly the result of several episodes of migration that began from the Ordovician to the end of the Cretaceous. The hypothesis best supported by the interpretation of geochemical analyses gives the brine a Devonian age of about 400 million years [9]. The brine from Gaspésie has been less studied and is less understood than the natural brine from the St. Lawrence Lowlands. It has been identified in oil wells and surface seeps for more than a century [10]. Geochemical studies allow us to postulate that the brine would be of Paleozoic marine origin, altered by prolonged interactions with rocks rich in potassium and rubidium, abundant from the middle of the Devonian in the region [11]. Locally and with varying levels of intensity, the initial brine shows signs of seawater dissolution by meteoritic water. In the Gaspé Basin, the presence of evaporites has been recognized in underlying Silurian units and in younger Devonian units in adjacent basins. It is very likely that its initial age is Devonian.

We used 16S rRNA amplicon sequencing to determine the identity and structure of the microbial communities living in the rock and groundwater of the deep aquifers from both the Bécancour and Gaspésie sites. Archaeal 16S rRNA gene sequencing was not successful, yielding only a small number of reads. Bacterial 16S rRNA gene sequencing allowed us to analyze community structure between sites originating from different geological eras and in different geological layers. We also carried out metagenomic reconstructions from one groundwater brine sample from the Bécancour site, 875 m deep, to delve into potential pathways used by these microbial communities to survive in the harsh conditions found in the deep biosphere, to which osmotic stress due to the high salinity is added.

## 2. Materials and Methods

### 2.1. Study Sites and Sampling

Study sites are located in two different sedimentary basins (Table 1). Sites A198, A246, MA158, and FA239 are in the Saint Lawrence Lowland basin, mainly deposited during the Cambrian and Ordovician periods, close to the city of Bécancour (QC, Canada). Zones of interest were composed of sandy dolomite (A198 and A246), arkosic sandstone (MA158), and limestone (FA239). Sites FC139 and FC145 belong to the Gaspé Devonian Basin, in the Gaspésie region of Quebec. The geological formations encountered vary in composition, from silty limestone to dolomitic coarse limestone. All studied sites are deep wells drilled for the production of natural gas, brine (A198, A246, MA158, and FA239), or petroleum (FC139 and FC145), carried out by Ressources Utica Inc., Quebec, QC, Canada.

Natural brine at FC139 was collected during a production test on isolated sections of the well and frozen until filtration. Sampling of groundwater took place in July 2018 for deep wells A198 and A246 in Bécancour. We collected 2 L of water for each well in sterilized polypropylene bottles (Nalgene, Rochester, NY, USA), transported on ice, and stored at 4 °C until filtration in the lab, which was carried out the same day as sampling. Because the wells were pressurized, there was no need to use a pump for water collection. Groundwater physico-chemical parameters were measured continuously at the opening of the well (YSI multiparameter probe, model 10102030, Yellow Springs, OH, USA), and we waited for the numbers to stabilize, signaling stagnant water was discarded, before sampling. For the Bécancour wells, pH, temperature, dissolved oxygen, and conductivity were measured. Filtration was carried out using a 0.2 µm polyethersulfone filter (Sartorius, Germany), and filters were subsequently stored at −20 °C.

The rock samples were collected in June 2018 at the Natural Resource Ministry warehouse (Quebec City, QC, Canada), where samples are stored after a well is drilled. The rock samples were transferred in autoclaved aluminum sheets and stored on ice during transport and at −20 °C upon return to the lab.

### 2.2. Geochemical and Geological Analyses

For wells A198 and A246, water samples were collected in gas-free glass bottles after filtration on a 0.45 µm polyethersulfone filter (Sarstedt, Numbrecht, Germany) to measure dissolved organic carbon (DOC). Samples were analyzed with an OI Analytical Aurora 1030 W TOC Analyzer (College Station, TX, USA) by using the persulfate oxidation method at the GRIL-UQAM laboratory.

Gaseous ^13^C and ^12^C isotopes and concentrations for CO_2_ and CH_4_ were determined using cavity ring-down spectroscopy (CRDS) technology with the Picarro G2201-i Analyzer coupled with an autosampler (SAM1812001, OpenAutoSampler). The original concentration and δ^13^C isotopic signature of CH_4_ in the water were calculated as a function of headspace values, headspace ratios, temperature, pressure, and the isotopic fractionation across the liquid–gas interface. Dissolved gases were extracted from triplicate 100 mL water samples into 40 mL headspaces via exchange across the liquid–gas interface after 2 min of vigorous shaking. The initial headspace was composed of hydrocarbon-free ultra-zero air (Praxair, Danbury, CT, USA), and final headspaces were transferred to pre-evacuated 12 mL vials (Exetainer, Labco, Ceredigion, UK) for transport to the lab. Isotopic values were corrected post-processing using a calibration curve of known isotopic standards (δ^13^C CH_4_ = −66.5‰, −38.3‰, −23.9‰, concentration = 2500 ppm, diluted to 250 ppm) supplied from Air Liquide (http://airliquide.ca/, accessed on 20 July 2023). Concentrations, if not already known prior to processing via UGGA or gas chromatography, were corrected post-processing using curves of external standards supplied by Praxair (http://www.praxair.ca/en-ca, accessed on 20 July 2023).

X-ray fluorescence (XRF) was used to determine the elemental composition of the rock samples using a portable analyzer with the Mining Plus mode (DELTA, Olympus, Tokyo, Japan).

### 2.3. DNA Extraction and Sequencing

DNA was extracted from the filters using the DNeasy Power Water Kit (Qiagen, Hilden, Germany) according to the manufacturer’s instructions. DNA was extracted from the rock samples following the protocol used in Lazar et al. [12], using the DNeasy PowerMax Soil kit (Qiagen, Hilden, Germany). Prior to the extractions, the rock samples were rinsed with sterile autoclaved MilliQ water. We used 10 g of crushed rock powder and added a concentrated phosphate buffer to the initial extraction step, as suggested by Direito et al. [13]. Because the DNA is eluted in 5 mL of buffer, we concentrated the DNA in 200 µL of buffer, as in Lazar et al. [12], using glycogen and PEG6000. A negative extraction control was carried out with the DNeasy PowerMax Soil Kit and sterile MilliQ water. All extracted DNA samples were stored at −20 °C until further use.

Amplicon sequencing was carried out at the CERMO-FC genomic platform (the Center for Excellence in Research on Orphan Disease—Foundation Courtois) at UQAM. Archaeal and bacterial 16S rRNA genes were amplified with the Polymerase UCP HiFidelity PCR Kit (Qiagen, Hilden, Germany). We used the A340F (5′-CCCTACGGGCYCCASCAG-3′, [14])–A915R (5′-GTGCTCCCCCGCCAATTCCT-3′, [15]) primer pair to amplify the archaeal 16S rRNA genes and the B341F (5′-CCTACGGGAGGCAGCAG-3′, [16])–B785R (5′GACTACCGGGGTATCTAATCC-3′, [17]) primer pair to amplify the bacterial 16S rRNA genes. PCR amplification was performed under the following conditions: denaturation at 98 °C for 30 s, annealing for 30 s (58 °C for archaea, 57 °C for bacteria), extension at 72 °C for 1 min, and final extension at 72 °C for 10 min. We used 40 cycles for the archaea and 35 for the bacteria. Sequencing was performed using an Illumina MiSeq 2300 and the MiSeq reagent kit v.3 (600 cycles, Illumina, San Diego, CA, USA). Negative controls for the PCR amplifications of both the bacteria and archaea were sequenced as well. All sequences were deposited on the National Center for Biotechnology Information platform (NCBI) under the BioProject ID PRJNA978621.

We had initially planned to carry out shotgun sequencing for all three deep groundwater samples. However, only well A246 yielded enough DNA for shotgun sequencing. The shotgun-type DNA library was prepared with the NEBNext^®^ Ultra™ II FS DNA Library Prep Kit for Illumina (New England BioLabs Inc., Ipswich, MA, USA) following the manufacturer’s protocol. Ten nanograms of DNA was used as the starting material for library preparation. The fragmentation step was optimized to obtain inserts of 100–250 bp size. A size selection step with AMPure XP beads (Beckman Coulter, Brea, CA, USA) was added after the adapter ligation. A Unique Dual Index Primer Pair (UDI) was used during the PCR enrichment step (New England BioLabs Inc.). Libraries were quantified using the Qubit™ dsDNA HS Assay Kit (Invitrogen, Waltham, MA, USA). The average size of the fragment was determined using a Bioanalyzer (Agilent) instrument. The library was sequenced with a depth of 50 M reads on NovaSeq 6000 S1 (v1.5): paired-end 100 pb. Library preparation was carried out at the CERMO-FC sequencing platform (UQAM), and sequencing was carried out at the Genomic Center (CHU Québec-Université Laval). All sequences were deposited on the National Center for Biotechnology Information platform (NCBI) under accession numbers PRJNA978621.

### 2.4. Amplicon Sequence Analyses

The obtained sequences were analyzed using the Mothur software v.1.44.3 [18] and classified using the SILVA database v.138.1 [19]. Because the SILVA database does not contain subgroups for several unclassified archaeal taxa, we further implemented our analysis with reference sequences from the *Bathyarchaeaota* [20] and the *Woesearchaeota* [21], as well as a personal database. Amplicon sequence variants (ASVs) were computed using Mothur. Rarefaction was carried out using the median sequencing depth method [22]. Before running rarefaction analyses, we subtracted the ASVs that were sequenced in the PCR negative control from all samples and from the kit negative control for the rock samples.

### 2.5. Shotgun Sequence Analyses

Raw reads were quality filtered and trimmed using Sickle v1.3 [23] and assembled with SPAdes v3.15.5 [24]. Scaffold coverage was generated by mapping the entire filtered read set to the assembled scaffolds with BWA-MEM v0.7.17 [25]. Scaffolds were clustered in metagenome-assembled genomes (MAGs) based on their tetranucleotide frequencies and read coverage using Metabat2 version v2.2.15 [26]. Only MAGs with completeness > 50% and contamination < 10% according to CheckM v1.2.2 [27] were selected for further analysis. The Genome Taxonomy Database Toolkit (GTDB-Tk) v1.4.1 [28] was used to assign taxonomy to these genomes. Gene calling and functional annotation of assembled scaffolds and MAGs were performed using DRAM v1.4.6 [29] with databases dbCAN [30], KOfam [31], Merops [32], Pfam [33], Uniref90 [34], and VOGdb (http://vogdb.org, accessed on 15 June 2023).

To identify viral sequences in assembled scaffolds, the PhaMer tool [35], available through the online server PhaBOX [36], was used with default parameters. Taxonomic classification of identified viral contigs at the family level was conducted using the semi-supervised learning model PhaGCN [37] with default parameters. VirSorter2 v2.2.3 [38], followed by DRAM, was used for annotation of the viral sequences.

### 2.6. Statistical Analyses Analyses

Shannon diversity indices (α-diversity) were calculated using Mothur and compared between groups using a Wilcoxon test with the wilcox.test function in R [39] v.4.1.2. Community composition (ß-diversity) was observed with principal coordinate analysis (PCoA), in Mothur, using rarefied ASV tables and a Bray–Curtis dissimilarity distance matrix.

We tested community composition variance linkage with environmental parameters using permutational multivariate analyses (PERMANOVA) on the rarefied ASV tables in R with the adonis function of the vegan package. Significantly different taxa (genus level) between sample groups were determined using linear discriminant effect size (LefSe) analyses with the online tool from the Huttenhower lab (https://huttenhower.sph.harvard.edu/lefse/, accessed on 15 June 2023). Unique and shared genera between surface and groundwater samples were computed in Mothur.

We used distance-based redundancy analysis (db-RDA) to determine whether rock composition had a significant impact on microbial community composition. ASV tables were transformed and used to calculate a Bray–Curtis dissimilarity distance matrix. Environmental variables were transformed using log(x + 1). The db-RDA was applied to the distance matrix and the set of explanatory variables with the capscale function of the vegan package in R, and the significance of explanatory variables was assessed with the anova function in R with 200 permutations. The unique and shared contributions of each significant variable were determined with the varpart function of the vegan package in R.

## 3. Results

### 3.1. Geochemical and Physico-Chemical Properties of the Hypersaline Groundwater

Groundwater from wells A198 and A246 contained extremely high amounts of methane with a light δ^13^C CH_4_ (Table 2). The pH was slightly alkaline in A246, and dissolved organic carbon and CO_2_ were higher than in A198. Pinti et al. [9] reported 94 to 315 g/L salinity for these aquifers, up to ten times more than seawater.

### 3.2. Microbial 16S rRNA Gene Diversity

Both the groundwater and rock samples yielded only a few reads for the *Archaea* (between 1 and 795, Appendix A). Therefore, it was impossible to carry out any statistical analyses on the archaeal communities. Ammonia oxidizers (cand. Nitrosotenuis, cand. Nitrososphaera, cand. Nitrosotalea, and *Nitrosoarchaeum*), *Woesearchaeota* (subgroups 5a, 5b, and 24), and methanogens (*Methanosaeta*, *Methanosarcina*, *Methanobacterium*, *Methanocella*, *Methanomassiliococcus*, and *Methanoregula*) were among the taxa identified in the deep hypersaline groundwater. The same ammonia oxidizers and methanogens were found in the rock samples as well as in *Bathyarchaeota* (subgroups 6 and 18).

The *Proteobacteria* dominated the groundwater and rock samples at the phylum level (31 to 60% of the total number of reads for the groundwater and 41 to 93% for the rock samples), apart from the A198 site, which was dominated by *Actinobacteriota* (48 to 84%) (Figure 1a). At the genus level, the groundwater samples contained taxa affiliated with *Staphylococcus* (9%), *Corynebacterium* (8 to 10%), *Sphingomonas* (1 to 8%), *Acinetobacter* (4 to 20%), and *Hyphomicrobium* (6%) (Figure 1b). The rock samples from Bécancour were dominated by *Conexibacter* (2 to 15%), *Nocardioides* (2 to 24%), *Acinetobacter* (1 to 25%), *Corynebacterium* (1 to 14%), *Bacillus* (1 to 24%), and *Pseudomonas* (1 to 54%); while the samples from Gaspésie were dominated by *Xanthomonas* (48 to 74%), *Lactococcus* (10 to 17%), *Alkalibacterium* (1 to 24%), *Pseudomonas* (39 to 49%), and *Acinetobacter* (17 to 24%).

### 3.3. Bacterial α-Diversity Indices and Comparison

For the groundwater samples, bacterial α-diversity indices ranged between 3.51–5, 1.91–4.92 for the rock samples from Bécancour, and 0.99–3.93 for the rock samples from Gaspésie (Figure 2). Bacterial richness indices ranged from 99–320 for the groundwater samples, 26–32,564 for the rock samples from Bécancour, and 523–105,586 for the rock samples from Gaspésie (Appendix A). Bacterial evenness indices ranged from 0.74–0.9 for the groundwater samples, 0.45–0.73 for the rock samples from Bécancour, and 0.19–0.54 for the rock samples from Gaspésie (Appendix A). We compared indices based on site (Bécancour or Gaspésie), and richness and evenness indices were significantly different and lower for the Bécancour sites, whether we included or not the groundwater samples (Table 3). When excluding the groundwater samples (only rock samples), Shannon diversity indices were significantly different and lower for the Gaspésie sites. Therefore, for the rock samples, the Bécancour sites had a higher diversity than the Gaspésie sites.

### 3.4. Bacterial Community Composition

The PCoA shows that the community composition of the Bécancour and Gaspésie rock samples clustered in different areas of the graph (Figure 3). Groups of rock samples based on geological layers also clustered apart, following well sites, apart from A198, which was composed of 7 different layers. This was confirmed with the PERMANOVA tests. When using all groundwater and rock samples, habitat (groundwater or rock) and site (Bécancour or Gaspésie) explained 4.6% and 14.8% of the bacterial variance, respectively (Table 4). When using only the rock samples, we observed that site and geological characteristics (layers) explained 9.4% and 17.3% of the bacterial variance, respectively. The significant difference between sites, excluding groundwater samples, was explained by *Exiguobacterium* and *Lysobacter,* which were significantly higher at Bécancour, and *Xanthomonas*, *Lactococcus*, *Alkalibacterium*, and *Trichococcus,* which were significantly higher at the Galt sites (Figure 4a). Finally, when looking only at the different geological layers, *Alkalibacterium* was significantly higher in the Forillon geological layers; *Pseudomonas*, *Acinetobacter*, *Rheinheimera*, and *Phenylobacterium* in the Indian Cove layers; *Exiguobacterium* in the Potsdam layers; and *Nocardioides* and *Limnobacter* in the Trenton layers (Figure 4b).

### 3.5. Shared and Unique Bacterial Genera

For the rock samples, the groups from Bécancour shared 30.72% of the genera with the group from Gaspésie, with 50.25% of the taxa being unique to Bécancour and 19.03 to Gaspésie (Appendix A). The dominant unique taxa in Bécancour were unc. *Microtrichales*, unc. *Gaielllales*, *Luteimonas*, *Marinilactibacillus*, *Mesorhizobium*, and *Iamia*. The dominant unique taxa in Gaspésie were *Lactococcus*, *Trichococcus*, *Lactobacillus*, *Enterobacter*, and unc. *Alteromonadaceae*.

For the rock samples, the groups from the Potsdam, Indian Cove, Forillon, and Trenton geological layers shared 6.84% of the genera, with 14.02% being unique to the Potsdam layer, 12.02% to the Indian Cove layer, 25.54% to the Trenton layer, and 6.18% to the Forillon layer (Appendix A). The dominant unique taxa in the Potsdam layers were *Luteimonas* and *Marinilactibacillus*; *Lysinimicrobium*, *Tabrizicola*, *Thalassospira*, and *Oxalicibacterium* in the Trenton layers; *Lactococcus* and *Lactobacillus* in the Forillon layers; and *Trichococcus*, *Sphingorhabdus*, *Solibacillus*, and *Herminiimonas* in the Indian Cove layers.

### 3.6. Rock Bacterial Community Correlation with Environmental Parameters

The correlation between bacterial community composition and environmental parameters was carried out with the rock samples and the XRF measurements. We conducted an initial db-RDA analysis using major cations (Mg, S, K, Ca) and major biological transition metals (V, Mn, Fe, Ni, Cu, Zn) [40]. Potassium was the only atom to significantly correlate with community composition (Table 5, Appendix A), explaining 2% of the bacterial variance. We also conducted a db-RDA analysis using atoms involved in specialized uses by some microbes (Al, Si, Ti, Cr, As, Rb, Sr, Y, and Pb). Aluminium and strontium were significantly correlated with community composition (Table 5, Appendix A), explaining 6 and 4.7% of the bacterial variance, respectively.

### 3.7. Metagenomic Assembly and Predicted Metabolic Pathways in the A246 Groundwater

A 1500 bp contig was assembled from the DNA extracted from the A246 groundwater, and 7 MAGs were constructed. Bins 2, 7, 9, and 11 were affiliated with *Sphingomonas* (which represented 1.67% of the bacterial 16S rRNA sequence reads), bin 1 with *Acinetobacter* (20.76%), bin 3 with *Massilia* (0.47%), and bin 4 with *Pseudomonas* (1.19%).

#### 3.7.1. Information Systems, and Anabolic Pathways

All MAGs contained genes involved in replication (DNA polymerase), transcription (tRNA and RNA polymerase), and translation (ribosome). Bins 1, 3, 4, 7, and 11 also contained an RNA polymerase (Appendix A). All MAGs contained genes encoding enzymes involved in nucleotide, major amino acids (Cys, His, Ile, Leu, Lys, Met, Orn, Phe, Pro, Ser, Thr, Trp, Tyr, Val), cobalamin, siroheme, heme, and coenzyme Q biosynthesis. In addition, bins 1 (*Acinetobacter*) and 4 (*Pseudomonas*) had genes encoding enzymes involved in betaine biosynthesis. Furthermore, all MAGs except bin 1 (*Acinetobacter*) had a set of genes for flagella biosynthesis and motor function.

#### 3.7.2. Transporters

All four *Sphingomonas*-affiliated MAGs contained genes encoding proteins involved in the transport of phosphate and iron, as well as an osmoprotectant transport system (Appendix A). In addition, bins 2, 9, and 11 contain molybdate and nickel transporters; bins 2 and 9 contain a putative multiple sugar transporter; and bins 7 and 11 contain a putative nitrate transporter. Bin 1 had a D-methionine transporter. The *Acinetobacter*-affiliated MAG contained the same D-methionine, molybdate, phosphate, and putative nitrate transporters. In addition, we found transporters for glutamate/aspartate, histidine, peptides, zinc, sulfate/thiosulfate, sulfonate, and urea. The *Massilia*-affiliated MAG contained the same molybdate, nickel, iron, phosphate, and putative nitrate transporters. In addition, we found transporters for glycine, betaine/proline, branched-chain amino acids, peptides, polar amino acids (putative), sodium, sulfate/thiosulfate, sulfonate, D-xylose, and rhamnose. The *Pseudomonas*-affiliated MAG contained the same osmoprotectant, molybdate, nickel, iron, and phosphate transporters. In addition, we found transporters for glycine betaine/proline, phosphonate, glycerol, inositol, L-cystine, arginine/ornithine, glutamate/aspartate, histidine, dipeptide, L-amino acid, branched-chain amino acid, peptides, polar amino acids (putative), zinc/manganese, sodium, nitrate, sulfate/thiosulfate, sulfonate, urea, D-xylose, glucose, mannose, L-arabinose, maltose, ribose, sorbitol/mannitol, xylitol (putative), putrescine, and taurine.

#### 3.7.3. Carbon Utilization

All MAGs contained genes encoding proteins involved in protein degradation: exopeptidases for extracellular breakdown, peptidases for intracellular breakdown, and enzymes involved in the degradation of several amino acids (Leu, Lys, His, Met, and Tyr) (Appendix A). Bin 4 (*Pseudomonas*) also had enzymes involved in pyrimidine degradation.

All MAGs also had genes encoding proteins involved in glycolysis (Embden–Meyerhoff, Entner–Doudoroff, and the semi-phosphorylative Entner–Doudoroff pathways) and gluconeogenesis, as well as saccharides (galactose, galacturonate, mannose, xylose, and sucrose), hydrocarbons (protocatechuate and 3-phenylpropionate), lignin (polyphenol oxidase), and fatty acids (malonate semialdehyde pathway) degradation. In addition, bin 9 had genes involved in xylene/toluene hydrocarbon degradation; bin 11 had genes involved in benzoate and catechol hydrocarbon degradation; bin 1 had genes involved in catechol, benzoate/toluene, and naphthalene hydrocarbon degradation; and bin 4 had genes involved in benzoate, aminobenzoate, naphthalene, and toluene hydrocarbon degradation. Furthermore, all MAGs also contained a wide number of CAZymes (carbohydrate-active enzymes) involved in α-galactan, mannan, cellulose, β-galactan, chitin, fucose, mucin, pectin, starch, xylan, and xyloglucan cleavage. The *Acinetobacter*-affiliated MAG contained fewer CAZymes, mainly involved in polyphenolic, cellulose, and chitin cleavage.

All MAGs had numerous pathways for carbon fixation: dicarboxylate-hydroxybutyrate, hydroxypropionate-hydroxybutylate, reductive acetyl-CoA, reductive citrate, reductive pentose phosphate, and reductive glycine cycles.

The pyruvate and acetyl-CoA produced in all these carbon degradation pathways have the potential to enter many central metabolism pathways. Indeed, all MAGs have genes encoding proteins involved in pyruvate metabolism, the TCA, and the glyoxylate cycles. Acetyl-CoA can also be converted to acetate, catalyzed by either an acetate kinase (producing ATP) or an acetyl-CoA synthetase. Apart from bin 7, ethanol fermentation can also occur, recycling NADH, catalyzed by alcohol dehydrogenases.

All MAGs contain genes encoding proteins involved in anabolic pathways such as the pentose phosphate, shikimate, and glucuronate (except bin 7) pathways, methylaspartate, and urea cycles.

#### 3.7.4. Energy Metabolism

All MAGs have genes encoding the subunits for an NADH-quinone oxidoreductase (complex I), the first complex of the respiratory chain, which pumps protons and recycles NADH [41,42], as well as an ATP synthase (Appendix A). For the electron acceptors, we also detected cytochromes bd, c, and o if oxygen is used, but also an arsenate reductase and a thiosulfate reductase in bin 3. We identified a sulfur-oxidizing protein in bin 2, with thiosulfate as a possible energy source.

All MAGs contained genes involved in sulfur and nitrogen assimilation. In addition, bins 1 and 4 had genes involved in assimilatory and dissimilatory nitrate reduction, and bin 3 had genes involved in dissimilatory nitrate reduction.

### 3.8. Bacteriophage Identification and Annotation in the A246 Groundwater

We identified 201 phages in the groundwater from well A246, and 151 had no determined taxonomic affiliation. All identified phages belonged to the *Caudovirales*, and the families that we identified were *Ackermannviridae* (1), *Casjensviridae* (2), *Drexlerviridae* (4), *Herelleviridae* (2), *Kyanoviridae* (3), *Mesyanzhinovviridae* (2), *Orlajensenviridae* (6), *Peduoviridae* (19), *Straboviridae* (2), and *Zierdtviridae* (10). The annotated viral genes encoded proteins involved in the dicarboxylate-hydroxybutyrate and hydroxypropionate-hydroxybutylate cycles, glucuronate, glycolysis, and gluconeogenesis pathways, and amino acid biosynthesis.

## 4. Discussion

### 4.1. Bacterial Community Diversity and Structure in the Deep Aquifers

#### 4.1.1. Differences between the Bécancour and Gaspésie Regions

Whether we included or not the groundwater samples in the analyses, we observed that bacterial richness and evenness indices were significantly lower for samples in Bécancour compared to Gaspésie. Similarly, at the β-community level, the site explained a notable percentage of the community variance. The Bécancour sites were formed 460 million years ago (Ordovician era), while the Gaspésie sites are younger, emerging 410 million years ago (Devonian era). Although no direct correlation can be established between site age and bacterial communities, environments were strikingly different throughout both of these geological eras. During the Ordovician, most of the planet was covered in water, and marine invertebrates and green and red algae were the only known living organisms [43]. The Devonian is marked by a terrestrial landscape and the presence of plants [43]. Therefore, abiotic conditions, nutrient sources, and carbon sources would have been drastically different at both sites and throughout both eras, potentially shaping very different bacterial communities.

When only working with the rock samples, we found that although the richness indices were lower in Bécancour, half the taxa were unique to this site, possibly because the site is older. *Exiguobacterium* and *Lysobacter* genera were significantly higher in Bécancour. *Lysobacter* was initially found in Canadian soils and lake water, but also in soil, plants, and freshwater systems. It is described as mucilaginous and able to survive in oligotrophic conditions by killing other bacteria in its habitat [44,45]. *Exiguobacterium* can grow on a wide range of abiotic parameters such as temperature, pH, or salinity and uses a variety of polysaccharides and proteins to survive [46,47]. Furthermore, the *Luteimonas*, *Marinilactibacillus*, *Mesorhizobium*, and *Iamia* genera were unique to the Bécancour site. All these genera have either been exclusively collected or have strains collected from marine waters (i.e., *Luteimonas* from algae or deep-sea sediments [48,49], *Marinilactibacillus* from marine organisms and deep subseafloor sediments [50,51], *Mesorhizobium* from marine habitats [52], and *Iamia* from a sea cucumber [53]). Thus, the detection of these taxa only at the Ordovician site might be linked to the marine characteristics of this era.

*Xanthomonas* and *Alkalibacterium* were significantly higher in Gaspésie. *Xanthomonas* is usually associated with plants, with pathogenic and non-pathogenic strains [54]. This bacterium is able to withstand desiccation, cold temperatures, oxidative, and osmotic stresses [55] and was found in dolomite rocks of the central Alps [56]. *Alkalibacterium* is a slightly halophilic lactic acid bacteria found in salted fish and saline soils [57,58]. All dominant and unique taxa in Gaspésie were also lactic acid bacteria (i.e., *Lactococcus* and *Lactobacillus* [59], and *Trichococcus* [60]). In addition, *Trichococcus* contains genes protecting it from low temperatures and osmotic stress [61]. Hence, in the Gaspésie samples, we identified fermenters and plant-associated bacteria, possibly linked to the presence of plants during the Devonian era.

More generally, we found in the endolithic community from both sites taxa with the potential to withstand harsh abiotic conditions, such as cold temperature, desiccation, or osmotic stress, probably explaining their dominance in these samples. Bacterial predation and fermentation could also be strategies to survive in rock habitats with few nutrient or carbon sources.

#### 4.1.2. Influence of the Abiotic Environment on the Endolithic Bacterial Community

The four tested geological layers (Potsdam, Indian Cove, Forillon, and Trenton) had a significant impact on and explained an important proportion of bacterial community variance. The Potsdam rock layers are characterized by interbedded sandstone and dolomite. It is a rock unit formed of sediments from unvegetated landscapes, consisting almost entirely of quartz [62]. The Trenton rocks are mainly calcilulite limestone (https://igws.indiana.edu/compendium/trenton-limestone, accessed on 20 July 2023) typical of marine and lacustrine environments, while the Forillon and Indian Cove formations are silty limestone [63,64].

The difference in the bacterial community of the Potsdam rock was explained mainly by *Exiguobacterium*, while unique taxa were *Luteimonas* and *Marinilactibacillus*. As mentioned in the previous paragraph, these bacterial genera are likely linked to the marine origin of the rock samples. *Exiguobacterium* is characterized as an extremophilic bacterium, able to withstand high salinity and also scavenge for nutrients such as phosphorous [65], which could be very advantageous in quartz habitats. Indeed, quartz is considered a nutrient-poor mineral that can be colonized by specific microbial phylotypes [66]. The Trenton layer community was explained by *Nocardioides* and *Limnobacter*, and *Lysinimicrobium*, *Tabrizicola*, *Thalassospira,* and *Oxalicibacterium* were unique taxa. *Nocardioides* is found in terrestrial and aquatic environments, as well as low-nutrient habitats such as oligotrophic cave rocks, marine sediments, or Siberian permafrost [67,68,69]. *Lysinimicrobium* and *Tabrizicola* are both tolerant to high salt concentrations [70,71], possible remnants of the marine origin of the rocks. *Limnobacter* and *Thalassospira* both exhibit metabolisms useful in low-nutrient oligotrophic conditions (thiosulfate oxidation [72], phosphate chemotaxis [73]). *Oxalicibacterium* is an oxalotrophic bacteria that uses oxaloacetate as its sole carbon source [74]. Calcium oxalate in limestone is used by microorganisms to grow [75,76] and could be an important source of carbon for endolithic bacteria.

The difference in the bacterial community of the Forillon rock was explained mainly by *Alkalibacterium*, while unique taxa were *Lactococcus* and *Lactobacillus*, all discussed in the previous paragraph. Finally, the Indian Cove layer community was explained by *Pseudomonas*, *Acinetobacter*, *Rheinheimera*, and *Phenylobacterium*, and *Trichococcus*, *Sphingorhabdus*, *Solibacillus*, and *Herminiimonas* were unique taxa. Many of these taxa are common soil bacteria [77,78,79,80,81], while *Rheinheimera* is found in mangrove sediments [82] and *Herminiimonas* in deep glacial ice [83] and mineral water [84]. These taxa could be indicative of the terrestrial and plant-covered land from which these rocks were formed.

Potassium, aluminium, and strontium were significantly correlated with bacterial community composition. Potassium, a major intracellular cation, can be freed through rock weathering, and potassium-bearing minerals can be solubilized by bacteria [85]. In contrast, the link between aluminium and strontium would mostly be tolerance mechanisms when facing rock environments with compounds toxic to bacterial cells [86,87,88,89].

### 4.2. Deep Hypersaline Groundwater Microbial Communities

#### 4.2.1. Groundwater Community Taxonomic Diversity

Extremely high concentrations of methane were detected in the groundwater of Bécancour wells A198 and A246 [90], with higher measured values in the A246 groundwater. δ^13^C CH_4_ values showed that this methane was majority thermogenic [91], suggesting that the identified methanogenic sequences played only a minor role in the community. Despite these high methane concentrations, we did not detect any known methane oxidizers in either the archaeal (anaerobic methane oxidizers) or bacterial (aerobic) datasets, and no functional genes were detected in the MAGs from A246. Most likely, the hypersaline characteristics of the water did not enable methane oxidizers to either be present or active [92]. Most of the detected archaeal sequences were affiliated with ammonia oxidizers and *Woesearchaeota*. The ammonia oxidizers identified in our study are not typical high-salt-environment dwellers, but *Woesearchaeota* have been found to survive in these types of habitats, potentially carrying out fermentation [93]. All dominant bacterial taxa, except *Hyphomicrobium*, have been found in other hypersaline habitats (*Staphyloccocus* [94,95], *Corynebacterium* [96], *Sphingomonas* [97], *Acinetobacter* [98], *Paracocus* [99]), although most are often associated with soil environments [77,95,100,101,102,103].

#### 4.2.2. Potential Metabolisms of the Deep Groundwater Bacteria Based on Assembled Genomes

The seven assembled genomes were affiliated with *Acinetobacter*, which was the most relatively abundant genus in the A246 16S rRNA sequence dataset, as well as *Sphingomonas*, *Massilia*, and *Pseudomonas*. All these taxa were identified in the amplicon sequences, but at relative abundances lower than 2%. Although they are living in very harsh conditions (i.e., oligotrophic, hypoxic, hypersaline, and dark) with no new fluids imported in the last 2 million years [9], these bacteria do not seem to have resorted to reducing their genomes, which is a stress coping mechanism [104], since they still possess pathways for the biosynthesis of amino acids, vitamins, co-factors, and nucleotides. They are also able to carry out essential functions such as replication, transcription, and translation. Apart from the *Acinetobacter* genome, all bacteria are mobile and have the potential to produce all components of a flagella. This would give the bacteria more opportunities to scavenge for nutrients, energy, and carbon sources.

There are mainly three potential carbon sources, supported by the presence of measurable dissolved organic carbon. The bacteria have the pathways to hunt and cut extracellular proteins/peptides into smaller molecules able to go through transporters that we also identified and to degrade these peptides down to glutamate, acetoacetyl-CoA, and acetyl-CoA. These extracellular peptides and amino acids are often found as detritus in marine sediments [105], and free amino acids can be found in groundwater [106], although not as deep as the one in our study. Moreover, free amino acids might increase in high-salt environments [107]. Secondly, the bacteria could degrade saccharides (mostly galactose), even though most genomes did not contain specific transporters, down to glucose being able to enter the glycolysis pathway or glyceraldehyde being able to enter the pentose phosphate pathway. They all contained a high number and diversity of CAZymes, indicating a potential capacity to degrade plant-derived complex organic molecules, such as galacturonate [108], galactan [109], mannan [110], cellulose, pectin [111], fucose [112], xylan [113], and xyloglucan [114], as well as fungus-derived complex organic molecules such as chitin [115]. If present, the source of these organic compounds is uncertain. Although the detection of these genes does not necessarily mean that these compounds are present, the presence of signatures of microorganisms associated with land plants in the Bécancour brine could be a significant discovery supporting the hypothesis of a terrestrial presence of evolved plants from the Cambrian–Ordovician.

Galactan, mainly found in red marine algae, could have been present in the original basin water, which later became entrapped in the subsurface. Red marine algae are known to be one of the oldest forms of life encountered on the planet. Gibson [116] dated the appearance of the earliest known expression of extant forms of multicellularity and eukaryotic photosynthesis to a little over 1000 million years ago. However, Morris et al. [117] suggest that land plants were present during the Ordovician and Cambrian, well-preserved geological eras in the St. Lawrence Lowlands. This timescale implies the early establishment of terrestrial ecosystems by terrestrial plants, advancing their appearance by more than 100 million years compared to standard fossil dating. Bottomley [118] suggests that the composition of the deep hypersaline brines is residual brine from a younger Paleozoic sedimentary basin. According to this theory, residual high-density brines, formed by the evaporation of seawater during the Devonian, whose geological units are now completely eroded, infiltrated the underlying Ordovician and Cambrian bedrock, where subsequent chemical changes occurred. Evaporites in the St. Lawrence Lowlands likely existed only during Devonian–Silurian time. Brines might result from the infiltration of Devonian water leaching halite, penetrating or below the deeper Cambrian–Ordovician aquifers [9]. These high-density brines could have also migrated at depth during the St. Lawrence rift, during Mesozoic time, from ocean water [119].

The bacteria contained enzymes involved in two glycolytic metabolisms: the Entner–Doudoroff and Embden–Meyerhof–Parnas pathways. The co-existence of both pathways has been described before in *Corynebacterium*, where it was shown to enhance glucose consumption [120]. The same was also observed in the halophilic bacterium *Chromobacter salexigens*, where it was suggested that the presence of both glycolytic pathways would be an advantage in hypersaline conditions when protection against the high salt concentrations comes at a high energetic cost [121]. Finally, the genomes contained pathways involved in hydrocarbon degradation, mostly benzoate but also aminobenzoate, naphthalene, or toluene, leading to pyruvate, acetyl-CoA, succinyl-CoA, or oxaloacetate. Although not measured, hydrocarbons were present in the groundwater [122], thereby explaining the presence of these pathways.

The bacteria are also possible autotrophs, based on the presence of several carbon fixation pathways and the detection of CO_2_. We detected genes involved in the dicarboxylate-hydroxybutyrate cycle, which was discovered in a hyperthermophilic archaeon [123]. This pathway combines CO_2_ fixation with the reduction of acetyl-CoA to pyruvate. Many of the enzymes involved in this pathway are also part of the reductive citrate cycle. This autotrophic pathway was shown to be less energy costly compared to other pathways such as the Calvin cycle and can only occur in strict anaerobic conditions. It is unclear whether oxygen is actually present in the groundwater since mixing could have occurred during sampling. We detected cytochromes, suggesting the bacteria use oxygen as the terminal electron acceptor, but we also detected arsenate, thiosulfate, and nitrate reductases, indicating the possibility of functioning under anaerobic conditions. Another potential CO_2_-fixation pathway is the reductive glycine pathway, which starts with the same enzymes as the reductive acetyl-CoA pathway and produces 5,10-methylene-THF (C1 module [124]). The glycine cleavage system then uses NH_3_ and CO_2_, leading to the production of glycine (C2 module). We did not detect the enzymes further degrading glycine to serine and ultimately pyruvate. This carbon fixation pathway only uses 1–2 ATP molecules [125] and is the most energy-efficient carbon fixation route. In extremely oligotrophic conditions such as those found in the deep hypersaline groundwater of Bécancour, this energy-saving pathway would be of considerable interest. All the intermediate compounds thus produced (e.g., pyruvate, acetyl-CoA, oxaloacetate) can then enter the central metabolism pathways we identified in the MAGs, such as the TCA and glyoxylate cycles, eventually leading to anabolic pathways.

We did not identify typically known halophilic bacteria, apart from *Pseudomonas,* which has one known halophilic genus [126]. This might be linked to the origin of the water and thus the surface microbial seedbank dating back more than 2 million years. The unique and enclosed conditions found within this groundwater could also explain this observation. However, the MAGs all contained enzymes or pathways allowing for adaptation and survival in hypersaline conditions. Mainly, these bacteria likely uptake or synthesize compatible solutes as a strategy since we detected genes encoding glutamate, proline, and glycine betaine transporters, as well as pathways involved in proline, glutamate, and betaine biosynthesis. In addition, the CO_2_-fixation pathway via the reductive glycine pathway probably serves as a glycine source. The high salt concentrations outside the bacterial cells lead to osmotic stress. The accumulation of these small molecules inside the cytoplasm allows it to counteract this osmotic pressure [127]. Glycine betaine is one of these compounds that offers the best osmoprotection, followed by proline and glutamate [128]. Finally, the *Massilia*-affiliated MAG also contained a sodium transport system, as well as an Na^+^/H^+^ antiporter, for maintaining Na^+^ homeostasis in high salt conditions [129].

#### 4.2.3. Groundwater Bacteriophages

All identified phages belonged to the *Caudovirales*, which are tailed phages that infect a wide range of bacterial hosts and are possibly as old as 3.8 billion years [130]. There are as yet very few studies on phage community diversity in groundwater habitats, although they most likely shape and have an impact on the diversity and structure of microbial communities. Two studies from saline habitats [131,132] also contained *Caudovirales* phages, but not the same families that we found in our samples. The authors suggested that high salinity did not decrease viral infection. This is likely the case in our hypersaline groundwater, given the high number of families that we detected. Hylling et al. [133] isolated two phages infecting *Bacillus* and *Pseudomonas* from groundwater samples. Kothari et al. [134] identified more viral taxa (including previously unknown taxa) and associated hosts in a groundwater ecosystem. They also detected metabolic genes in the phages, encoding enzymes involved in tolerance mechanisms, probably benefiting the infected bacterial hosts. Bhattari et al. [135] also found auxiliary metabolic genes in phages from the Great Salt Lake for carbon fixation or formaldehyde assimilation. The auxiliary metabolic genes we detected in our phages would give an advantage to the infected bacterial hosts as well, potentially helping them scavenge for carbon sources and protecting them from the hypersaline conditions.

## 5. Conclusions

Both bacterial diversity indices and community composition were significantly different between the Bécancour sites dating back to the marine-dominated Ordovician geological era and the Gaspésie sites going back to the plant-covered terrestrial landscapes of the Devonian era. The bacterial genera that explained these differences seem to reflect this age and origin difference, as most taxa from the Bécancour sites are detected in marine waters, while many taxa from the Gaspésie sites are common soil bacteria. Overall, many bacteria from the rock samples were affiliated with genera able to withstand cold temperature, desiccation, or osmotic stress and also with different strategies to survive in oligotrophic settings. The rock geological layers also had a significant impact on bacterial community composition. We observed a potential link with the geological era during which the rocks were formed, as well as bacterial taxa affiliated with genera able to scavenge for nutrients or carbon sources.

The deep hypersaline groundwater samples did not harbor typical halophilic archaea or bacteria, which could be attributed to the old age of the water. The genomes that were reconstructed from one groundwater sample show bacteria with the potential to biosynthesize their own essential molecules and to use proteins, plant-derived complex organic compounds, hydrocarbons, or inorganic carbon as carbon sources. Although none of the detected genera are common halophilic lineages, we detected typical strategies to cope with osmotic stress, such as the use of compatible solutes or the transport of sodium inside the cell. These genomes also encoded viral (bacteriophage) genes, highlighting a high community diversity, as well as auxiliary metabolic genes that might confer a selective advantage to the infected bacterial hosts. Therefore, despite the harsh conditions found in the deep hypersaline aquifers, both aquatic and endolithic bacteria might have the metabolic capabilities to survive and grow.

## Figures and Tables

**Figure 1 genes-14-01529-f001:**
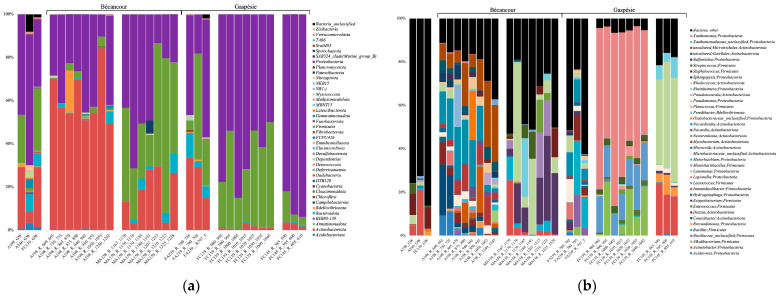
Bacterial 16S rRNA gene taxonomy at the phylum (**a**), and genus (**b**) levels for the groundwater and rock samples from the Bécancour and Gaspésie sites. GW, groundwater; R, rock.

**Figure 2 genes-14-01529-f002:**
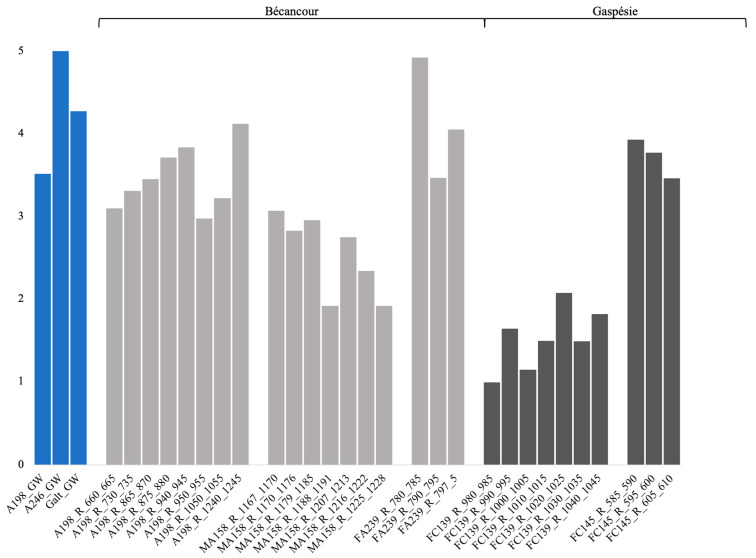
Bacterial Shannon diversity indices for the groundwater and rock samples from the Bécancour and Gaspésie sites. Light grey represents the Bécancour samples, and dark grey the Gaspésie samples. GW, groundwater; R, rock.

**Figure 3 genes-14-01529-f003:**
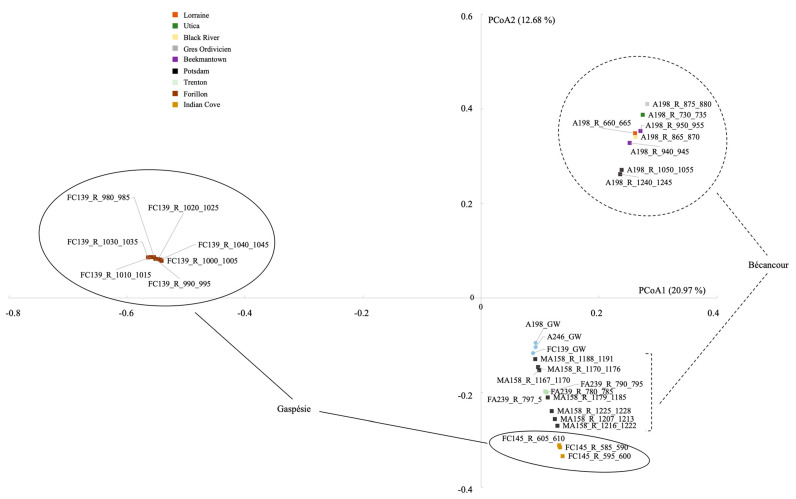
PCoA plot ordination showing distances between groundwater and rock bacterial communities, in both the Bécancour and Gaspésie sites, based on a Bray–Curtis matrix. GW, groundwater; R, rock.

**Figure 4 genes-14-01529-f004:**
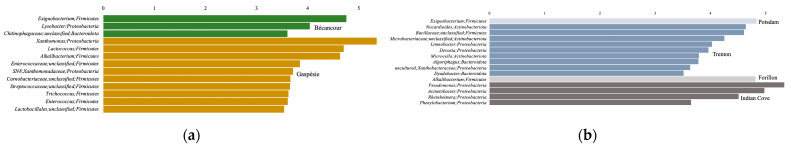
LEfSe analysis of the bacterial genera explaining differences in the community structure between the Bécancour and Gaspésie sites (**a**), or between the different geological layers (**b**). Only genera with an LDA value > 3.5 are shown.

**Table 1 genes-14-01529-t001:** List of samples used in this study, from the Bécancour and Gaspésie sites. mbs, meters below surface.

Sample Name	Depth (mbs)	Geological Layer
Bécancour A198 rock	660–665	Lorraine
	730–735	Utica
	865–870	Black River
	875–880	Grès Ordovicien
	940–945	Beekmantown
	950–955	Beekmantown
	1050–1055	Potsdam
	1240–1245	Potsdam
Bécancour A198 groundwater	937–955	Beekmantown
Bécancour A246 groundwater	875	Black River
Bécancour MA158 rock	1167–1170	Potsdam
	1170–1176	Potsdam
	1179–1185	Potsdam
	1188–1191	Potsdam
	1207–1213	Potsdam
	1216–1222	Potsdam
	1225–1228	Potsdam
Bécancour FA239 rock	780–785	Trenton
	790–795	Trenton
	797.5	Trenton
Gaspésie FC139 rock	980–985	Forillon
	990–995	Forillon
	1000–1005	Forillon
	1010–1015	Forillon
	1020–1025	Forillon
	1030–1035	Forillon
	1040–1045	Forillon
Gaspésie FC139 groundwater	1002	Forillon
Gaspésie FC145 rock	585–590	Indian Cove
	595–600	Indian Cove
	605–610	Indian Cove

**Table 2 genes-14-01529-t002:** Geochemical measurements made in the groundwater of wells A198 and A246 from the Bécancour site.

	A198	A246
pH	6.01	8.45
Temperature (°C)	17.5	25.6
Dissolved oxygen (%)	13.4	15.5
Dissolved organic carbon (mg/L)	0.09	16.2
CO_2_ (ppm)	616.97	858.36
CH4 (ppm)	189,148.77	1,198,300.5
δ^13^C CH_4_ (permil)	−39.74	−35.097

**Table 3 genes-14-01529-t003:** Pvalues for the diversity indices’ comparison between samples from the Bécancour and Gaspésie sites.

	Shannon	Chao1	Evenness
All samples	0.05389	0.02076	0.0002886
Rocks	0.03583	0.04089	3 × 10^−4^

**Table 4 genes-14-01529-t004:** PERMANOVA analysis of bacterial community composition explained by habitat (rock and groundwater), site (Bécancour and Gaspésie), or geological layers (Potsdam, Forillon, Trenton and Indian Cove).

All Samples					
	Df	SumOfSqs	R2	F	Pr(>F)
Habitat	1	0.6373	0.04651	1.6170	0.038
Site	1	2.0283	0.14804	5.1463	0.001
Residual	28	11.0355	0.80545		
Total	30	13.7010	1.00000		
**Rock samples**					
	**Df**	**SumOfSqs**	**R2**	**F**	**Pr(>F)**
Habitat	1	0.8697	0.09383	2.3047	0.022
Geology	2	1.6067	0.17335	2.1289	0.002
Residual	18	6.7922	0.73282		
Total	21	9.2686	1.00000		

**Table 5 genes-14-01529-t005:** Distance-based RDA for the rock samples showing correlation between bacterial composition and atoms composing the rocks.

Major Cations and Biological Transition Metals					
	Df	SumOfSqs	R2	F	Pr(>F)
Mg	1	0.4238	0.9852	0.464	0.464
S	1	0.5869	1.3644	0.091	0.091
K	1	0.7321	1.7018	0.024	0.024
Ca	1	0.4033	0.9375	0.561	0.561
V	1	0.4341	1.0091	0.395	0.395
Mn	1	0.3992	0.9279	0.518	0.518
Fe	1	0.5528	1.2851	0.144	0.144
Ni	1	0.4296	0.9987	0.445	0.445
Cu	1	0.5852	1.3604	0.095	0.095
Zn	1	0.3560	0.8275	0.743	0.743
Residual	14	6.0226			
**Specialized uses**					
	**Df**	**SumOfSqs**	**R2**	**F**	**Pr(>F)**
Al	1	0.8212	2.0807	0.005	0.005
Si	1	0.4860	1.2315	0.165	0.165
Ti	1	0.5320	1.3481	0.096	0.096
Cr	1	0.5855	1.4837	0.062	0.062
As	1	0.4826	1.2228	0.154	0.154
Rb	1	0.3459	0.8766	0.679	0.679
Sr	1	0.8256	2.0919	0.002	0.002
Y	1	0.4166	1.0557	0.364	0.364
Pb	1	0.5104	1.2933	0.120	0.120
Residual	15	5.9197			

## Data Availability

All sequences were deposited on the National Center for Biotechnology Information platform (NCBI) under the BioProject ID PRJNA978621.

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
