# Peer review of "Deep Isolated Aquifer Brines Harbor Atypical Halophilic Microbial Communities in Quebec, Canada"

_genes, 2023, doi:10.3390/genes14081529_

Round 1
Reviewer 1 Report
This manuscript is dedicated to the study of microbial diversity and microbial community structure in hypersaline habitats and examines the metabolic potential of these communities. The samples were collected at two locations at different depths and belonged to different geological layers. Although sequencing revealed atypical representatives of hypersaline habitats, the authors succeeded in adequately explaining these results. The text is well written. In addition, I would like to highlight the full and interesting discussion of the results. Nevertheless, here are some recommendations for authors to improve the manuscript.
Title of Section 3.2. "Microbial gene diversity" does not quite correctly reflect its content. This section describes the diversity and abundance of microorganisms according to the results of sequencing the hypervariable regions of only the 16S rRNA gene and no other genes. Therefore, the title of the section should be more accurate.
In the same section, the percentage of taxa presence should be specified in parentheses. Section "3.3. Bacterial α-diversity indices and their comparison" requires a short conclusion at the end.
Author Response
This manuscript is dedicated to the study of microbial diversity and microbial community structure in hypersaline habitats and examines the metabolic potential of these communities. The samples were collected at two locations at different depths and belonged to different geological layers. Although sequencing revealed atypical representatives of hypersaline habitats, the authors succeeded in adequately explaining these results. The text is well written. In addition, I would like to highlight the full and interesting discussion of the results. Nevertheless, here are some recommendations for authors to improve the manuscript.
Reply from author: We would like to thank the reviewer for taking the time to read and comment our manuscript. Please note that the accession number for the shotgun sequencing is now added to the text L189.
Title of Section 3.2. "Microbial gene diversity" does not quite correctly reflect its content. This section describes the diversity and abundance of microorganisms according to the results of sequencing the hypervariable regions of only the 16S rRNA gene and no other genes. Therefore, the title of the section should be more accurate.
Reply from author: We agree with the reviewer, we have therefore changed the title of the section to ‘Microbial 16S rRNA gene diversity’, L248.
In the same section, the percentage of taxa presence should be specified in parentheses.
Reply from author: For the Archaea we did not add percentage given the low number of sequences. However, the information was added for the Bacteria L258-266.
Section "3.3. Bacterial α-diversity indices and their comparison" requires a short conclusion at the end.
Reply from author: We added a sentence L282-283.
Reviewer 2 Report
The MS is suitable for publication in its present form.
Author Response
We would like to thank the reviewer for taking the time to read and comment our manuscript. Please note that the accession number for the shotgun sequencing is now added to the text L189.Reviewer 3 Report
This manuscript reported the halophilic microbial community analysis of rock and groundwater from deep, isolated brine aquifers in 2 regions 19 dating back to the Ordovician and Devonian. The result would provide new information for favoring deep waste deposits. The manuscript was well written and organized. Some minor point should be improved.
- Please refer to the following “Nomenclature of Microorganisms” to revise throughout the manuscript.
"Names of all bacterial taxa (kingdoms, phyla, classes, orders, families, genera, species, and subspecies) are printed in italics and should be italicized in the manuscript; strain designations and numbers are not."
https://jb.asm.org/content/nomenclature
- Fig 1,2,3,4 should be on a larger scale. The current scale is not readable. Also, please revise bacterial names (should be italicized).
- Line 254: it should be Fig. 1
Author Response
This manuscript reported the halophilic microbial community analysis of rock and groundwater from deep, isolated brine aquifers in 2 regions 19 dating back to the Ordovician and Devonian. The result would provide new information for favoring deep waste deposits. The manuscript was well written and organized. Some minor point should be improved.
Reply from author: We would like to thank the reviewer for taking the time to read and comment our manuscript. Please note that the accession number for the shotgun sequencing is now added to the text L189.
- Please refer to the following “Nomenclature of Microorganisms” to revise throughout the manuscript. "Names of all bacterial taxa (kingdoms, phyla, classes, orders, families, genera, species, and subspecies) are printed in italics and should be italicized in the manuscript; strain designations and numbers are not."
https://jb.asm.org/content/nomenclature
Reply from author: Bacterial and archaeal taxa were italicized in the manuscript, throughout the text.
- Fig 1,2,3,4 should be on a larger scale. The current scale is not readable. Also, please revise bacterial names (should be italicized).
Reply from author: Because of the word manuscript size threshold required for this journal, we had to lower the quality of the figures in order to upload the text during submission. However, we also uploaded higher quality figures separately, I’m sure the editors can give the reviewer access to these figures. Figures 1 and 4 were modified accordingly.
- Line 254: it should be Fig. 1
Reply from author: This was changed L268.